# A Resizable Mini-batch Gradient Descent based on a Multi-Armed Bandit

## Abstract

Determining the appropriate batch size for mini-batch gradient descent is always time consuming as it often relies on grid search. This paper considers a resizable mini-batch gradient descent (RMGD) algorithm based on a multi-armed bandit that achieves performance equivalent to that of best fixed batch-size. At each epoch, the RMGD samples a batch size according to a certain probability distribution proportional to a batch being successful in reducing the loss function. Sampling from this probability provides a mechanism for exploring different batch size and exploiting batch sizes with history of success. After obtaining the validation loss at each epoch with the sampled batch size, the probability distribution is updated to incorporate the effectiveness of the sampled batch size. Experimental results show that the RMGD achieves performance better than the best performing single batch size. It is surprising that the RMGD achieves better performance than grid search. Furthermore, it attains this performance in a shorter amount of time than grid search.

## 1 Introduction

Gradient descent (GD) is a common optimization algorithm for finding the minimum of the expected loss. It takes iterative steps proportional to the negative gradient of the loss function at each iteration. It is based on the observation that if the multi-variable loss functions $f(\boldsymbol{w})$ is differentiable at point $\boldsymbol{w}$, then $f(\boldsymbol{w})$ decreases fastest in the direction of the negative gradient of $f$ at $\boldsymbol{w}$, i.e., $-\nabla f(\boldsymbol{w})$. The model parameters are updated iteratively in GD as follows:

$$\boldsymbol{w}_{t+1} = \boldsymbol{w}_t - \eta_t \boldsymbol{g}_t, \qquad \boldsymbol{g}_t = \nabla_{\boldsymbol{w}} f(\boldsymbol{w}_t)$$

where $\boldsymbol{w}_t$, $\boldsymbol{g}_t$, and $\eta_t$ are the model parameters, gradients of $f$ with respect to $\boldsymbol{w}$, and learning rate at time $t$ respectively. For small enough $\eta_t$, $f(\boldsymbol{w}_t) \geq f(\boldsymbol{w}_{t+1})$ and ultimately the sequence of $\boldsymbol{w}_t$ will move down toward a local minimum. For a convex loss function, GD is guaranteed to converge to a global minimum with an appropriate learning rate.

There are various issues to consider in gradient-based optimization. First, GD can be extremely slow and impractical for large dataset: gradients of all the data have to be evaluated for each iteration. With larger data size, the convergence rate, the computational cost and memory become critical, and special care is required to minimize these factors. Second, for non-convex function which is often encountered in deep learning, GD can get stuck in a local minimum without the hope of escaping. Third, stochastic gradient descent (SGD), which is based on the gradient of a single training sample, has large gradient variance, and it requires a large number of iterations. This ultimately translates to slow convergence. Mini-batch gradient descent (MGD), which is based on the gradient over a small batch of training data, trades off between the robustness of SGD and the stability of GD. There are three advantages for using MGD over GD and SGD: 1) The batching allows both the efficiency of memory usage and implementations; 2) The model update frequency is higher than GD which allows for a more robust convergence avoiding local minimum; 3) MGD requires less iteration per epoch and provides a more stable update than SGD. For these reasons, MGD has been a popular algorithm for machine learning. However, selecting an appropriate batch size is difficult. Various studies suggest that there is a close link between performance and batch size used in MGD Breuel (2015); Keskar et al. (2016); Wilson & Martinez (2003).

There are various guidelines for selecting a batch size but have not been completely practical Bengio (2012). Grid search is a popular method but it comes at the expense of search time. There are a small number of adaptive MGD algorithms to replace grid search Byrd et al. (2012); De et al. (2016); Friedlander & Schmidt (2012). These algorithms increase the batch size gradually according to their own criterion. However, these algorithms are based on convex loss function and hard to be applied to deep learning. For non-convex optimization, it is difficult to determine the optimal batch size for best performance.

This paper considers a resizable mini-batch gradient descent (RMGD) algorithm based on a multi-armed bandit for achieving best performance in grid search by selecting an appropriate batch size at each epoch with a probability defined as a function of its previous success/failure. At each epoch, RMGD samples a batch size from its probability distribution, then uses the selected batch size for mini-batch gradient descent. After obtaining the validation loss at each epoch, the probability distribution is updated to incorporate the effectiveness of the sampled batch size. The benefit of RMGD is that it avoids the need for cumbersome grid search to achieve best performance and that it is simple enough to apply to any optimization algorithm using MGD. The detailed algorithm of RMGD are described in Section 4, and experimental results are presented in Section 5.

## 2 RELATED WORKS

There are only a few published results on the topic of batch size. It was empirically shown that SGD converged faster than GD on a large speech recognition database Wilson & Martinez (2003). It was determined that the range of learning rate resulting in low test errors was considerably getting smaller as the batch size increased on convolutional neural networks and that small batch size yielded the best test error, while large batch size could not yield comparable low error rate Breuel (2015). It was observed that larger batch size are more liable to converge to a sharp local minimum thus leading to poor generalization Keskar et al. (2016). It was found that the learning rate and the batch size controlled the trade-off between the depth and width of the minima in MGD Jastrzkebski et al. (2017).

A small number of adaptive MGD algorithms have been proposed. Byrd et al. (2012) introduced a methodology for using varying sample size in MGD. A relatively small batch size is chosen at the start, then the algorithm chooses a larger batch size when the optimization step does not produce improvement in the target objective function. They assumed that using a small batch size allowed rapid progress in the early stages, while a larger batch size yielded high accuracy. However, this assumption did not corresponded with later researches that reported the degradation of performance with large batch size Breuel (2015); Keskar et al. (2016); Mishkin et al. (2017). Another similar adaptive algorithm, which increases the batch size gradually as the iteration proceeded, was done by Friedlander & Schmidt (2012). The algorithm uses relatively few samples to approximate the gradient, and gradually increase the number of samples with a constant learning rate. It was observed that increasing the batch size is more effective than decaying the learning rate for reducing the number of iterations Smith et al. (2017). However, these increasing batch size algorithms lack flexibility since it is unidirectional. Balles et al. (2017) proposed a dynamic batch size adaptation algorithm. It estimates the variance of the stochastic gradients and adapts the batch size to decrease the variance. However, this algorithm needs to find the gradient variance and its computation depends on the number of model parameters.

Batch size can also be considered as a hyperparameter, and there have been some proposals based on bandit-based hyperparameter (but not batch size) optimization which maybe applicable for determining the best fixed batch size. Jamieson & Talwalkar (2016) introduced a successive halving algorithm. This algorithm uniformly allocates a budget to a set of hyperparameter configurations, evaluates the performance of all configurations, and throws out the worst half until one configuration remains. Li et al. (2017) introduced a novel bandit-based hyperparameter optimization algorithm referred as HYPERBAND. This algorithm considers the optimization problem as a resource allocation problem. The two algorithms mentioned above are not adaptive, and for searching a small hyperparameter space, the two algorithms will not be very effective. The experimental results in this paper show that adaptive MGD tends to perform better than fixed MGD.

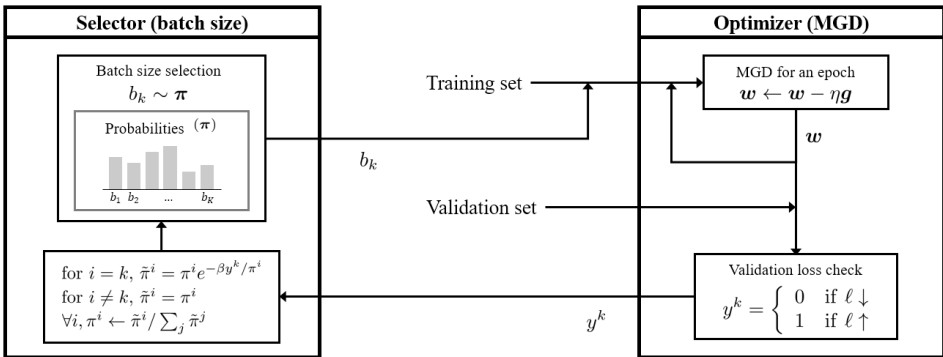

Figure 1: An overall framework of considered resizable mini-batch gradient descent algorithm (RMGD). The RMGD samples a batch size from a probability distribution, and parameters are updated by mini-batch gradient using the selected batch size. Then the probability distribution is updated by checking the validation loss.

## 3 SETUP

Let $\mathcal{B} = \{b_k\}_{k=1}^{K}$ be the set of possible batch size and $\boldsymbol{\pi} = \{\pi^k\}_{k=1}^{K}$ be the probability distribution of batch size where $b_k$, $\pi^k$, and $K$ are the $k^{\text{th}}$ batch size, the probability of $b_k$ to be selected, and number of batch sizes respectively. This paper considers algorithm for multi-armed bandit over $\mathcal{B}$ according to Algorithm 1. Let $\boldsymbol{w}_\tau \in \mathcal{W}$ be the model parameters at epoch $\tau$, and $\tilde{w}_t$ be the temporal parameters at sub iteration $t$. Let $J : \mathcal{W} \to \mathbb{R}$ be the training loss function and let $\boldsymbol{g} = \nabla J(\boldsymbol{w})$ be the gradients of training loss function with respect to the model parameters. $\eta_\tau$ is the learning rate at epoch $\tau$. Let $\ell : \mathcal{W} \to \mathbb{R}$ be the validation loss function, and $y^k \in \{0, 1\}$ be the cost of choosing the batch size $b_k$. In here, $y^k = 0$ if the validation loss decreases by the selected batch size $b_k$ (well-updating) and $y^k = 1$ otherwise (misupdating). The aim of the algorithm is to have low misupdating. For the cost function $y^k$, graduated losses such as hinge loss and percentage of non-negative changes in validation loss can be variations of 0-1 loss. However, there are no differences in regret bound among them in this setting and it is experimentally confirmed that there are little performance gaps among them. Therefore, this paper introduces the 0-1 loss, which is simple and basic.

## 4 RESIZABLE MINI-BATCH GRADIENT DESCENT

The resizable mini-batch gradient descent (RMGD) sets the batch sizes as multi arms, and at each epoch it samples one of the batch sizes from probability distribution. Then, it suffers a cost of selecting this batch size. Using the cost, probability distribution is updated.

### 4.1 ALGORITHMS

The overall framework of the RMGD algorithm is shown in Figure 1. The RMGD consists of two components: batch size selector and parameter optimizer. The selector samples a batch size from probability distribution and updates the distribution. The optimizer is usual mini-batch gradient.

**Selector** samples a batch size $b_{k_\tau} \in \mathcal{B}$ from the probability distribution $\boldsymbol{\pi}_\tau$ at each epoch $\tau$ where $k_\tau$ is selected index. Here $b_k$ is associated with probability $\pi^k$. The selected batch size $b_{k_\tau}$ is applied to optimizer for MGD at each epoch, and the selector gets cost $y^{k_\tau}$ from optimizer. Then, the selector

---

**Algorithm 1** Resizable Mini-batch Gradient Descent

---

**Input**:
$\mathcal{B} = \{b_k\}_{k=1}^{K}$ : Set of batch sizes
$\boldsymbol{\pi}_0 = \{1/K, \ldots, 1/K\}$ : Prior probability distribution

**Procedure**:
 1: Initialize model parameters $\boldsymbol{w}_0$
 2: **for** epoch $\tau = 0, 1, 2, \ldots$
 3:     Select batch size $b_{k_\tau} \in \mathcal{B}$ from $\boldsymbol{\pi}_\tau$
 4:     Set temporal parameters $\tilde{\boldsymbol{w}}_0 = \boldsymbol{w}_\tau$
 5:     **for** $t = 0, 1, \ldots, T - 1$ where $T = \lceil m/b_{k_\tau} \rceil$
 6:         Compute gradient $\boldsymbol{g}_t = \nabla J(\tilde{\boldsymbol{w}}_t)$
 7:         Update $\tilde{\boldsymbol{w}}_{t+1} = \tilde{\boldsymbol{w}}_t - \eta_\tau \boldsymbol{g}_t$
 8:     **end for**
 9:     Update $\boldsymbol{w}_{\tau+1} = \tilde{\boldsymbol{w}}_T$
10:     Observe validation loss $\ell(\boldsymbol{w}_{\tau+1})$
11:     **if** $\ell(\boldsymbol{w}_{\tau+1}) < \ell(\boldsymbol{w}_\tau)$
12:         Get cost $y^{k_\tau} = 0$
13:     **else**
14:         Get cost $y^{k_\tau} = 1$
15:     **end if**
16:     **for** $i = 1, 2, \ldots, K$
17:         **if** $i = k_\tau$
18:             Set temporal probability $\tilde{\pi}^i = \pi_\tau^i e^{-\beta y^{k_\tau}/\pi_\tau^i}$
19:         **else**
20:             Set temporal probability $\tilde{\pi}^i = \pi_\tau^i$
21:         **end if**
22:     **end for**
23:     Update $\forall i \in [K], \pi_{\tau+1}^i = \tilde{\pi}^i / \sum_j \tilde{\pi}^j$
24: **end for**

---

updates probabilities by randomized weighted majority,

$$\text{for } i = k_\tau, \quad \tilde{\pi}^i = \pi_\tau^i e^{-\beta y^{k_\tau}/\pi_\tau^i}$$
$$\text{for } i \neq k_\tau, \quad \tilde{\pi}^i = \pi_\tau^i$$
$$\forall i, \quad \pi_{\tau+1}^i = \tilde{\pi}^i / \sum_j \tilde{\pi}^j$$

where $\beta \in (0, 1)$ is positive hyperparameter. When $\tau = 0$, $\boldsymbol{\pi}_\tau = \{1/K, \ldots, 1/K\}$.

**Optimizer** updates the model parameters $\boldsymbol{w}$. For each epoch, temporal parameters $\tilde{\boldsymbol{w}}_0$ is set to $\boldsymbol{w}_\tau$, and MGD iterates $T = \lceil m/b_{k_\tau} \rceil^1$ times using the selected batch size $b_{k_\tau}$ where $m$ is the total number of training samples:

$$\tilde{\boldsymbol{w}}_{t+1} = \tilde{\boldsymbol{w}}_t - \eta_\tau \boldsymbol{g}_t, \quad \boldsymbol{g}_t = \nabla J(\tilde{\boldsymbol{w}}_t).$$

After $T$ iterations at epoch $\tau$, the model parameters is updated as $\boldsymbol{w}_{\tau+1} = \tilde{\boldsymbol{w}}_T$. Then, the optimizer obtains validation loss $\ell$, and outputs cost as follows:

$$y^{k_\tau} = \begin{cases} 0 & \text{if } \ell(\boldsymbol{w}_{\tau+1}) < \ell(\boldsymbol{w}_\tau) \\ 1 & \text{otherwise} \end{cases}.$$

The RMGD samples an appropriate batch size from a probability distribution at each epoch. This probability distribution encourages exploration of different batch size and then later exploits batch size with history of success, which means decreasing validation loss. Figure 2 shows an example of training progress of RMGD. The figure represents the probability distribution with respect to epoch. The white dot represents the selected batch size at each epoch. In the early stage of training,

---

[1] $\lceil x \rceil$ is the least integer that is greater than or equal to $x$

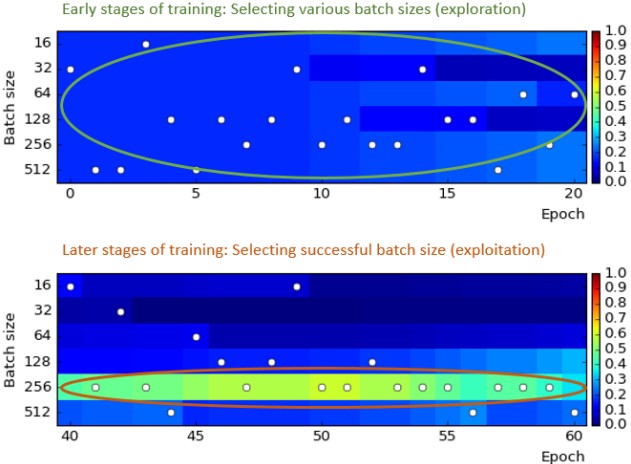

Figure 2: The probability distribution vs epoch using the RMGD. (top) The early stages of the training. (bottom) The later stages of the training. The white dot represents the selected batch size at each epoch. In the early stages of the training, RMGD updates the probabilities to search various batch sizes (exploration), and in the later stages, RMGD increases the probability of successful batch size (exploitation).

commonly, all batch sizes tend to decrease validation loss: $\pi$ is uniform. Thus, all batch size have equal probability of being sampled (*exploration*). In the later stages of training, the probability distribution varies based on success and failure. Thus, better performing batch size gets higher probability to be sampled (*exploitation*). In this case, 256 is the best performing batch size.

## 4.2 REGRET BOUND

The regret bound of the RMGD follows the regret bound derived in Shalev-Shwartz et al. (2012). The goal of this algorithm is to have low regret for not selecting the best performing batch size such that

$$\text{Regret}_{\mathcal{T}}(S) = \mathbb{E}\left[\sum_{\tau=1}^{\mathcal{T}} y_\tau^{k_\tau}\right] - \min_i \sum_{\tau=1}^{\mathcal{T}} y_\tau^i \tag{1}$$

where the expectation is over the algorithm's randomness of batch size selection and the second term on the right-hand side is the cumulative sum of the cost by the best fixed batch size which minimizes the cumulative sum of the cost. The regret of the RMGD is bounded,

$$\mathbb{E}\left[\sum_{\tau=1}^{\mathcal{T}} y_\tau^{k_\tau}\right] - \min_i \sum_{\tau=1}^{\mathcal{T}} y_\tau^i \leq \frac{\log K}{\beta} + \beta K \mathcal{T}. \tag{2}$$

In particular, setting $\beta = \sqrt{\log(K)/(K\mathcal{T})}$, the regret is bounded by $2\sqrt{K \log(K)\mathcal{T}}$, which is sublinear with $\mathcal{T}$. The detailed derivation of regret bound is described in the appendix A.

## 5 EXPERIMENTS

This section describes various experimental results on MNIST, CIFAR10, and CIFAR100 dataset. In the experiments, simple convolutional neural networks (CNN) is used for MNIST and 'All-CNN-C' Springenberg et al. (2014) is used for CIFAR10 and CIFAR100. The details of the dataset and experimental settings are presented in the appendix B.

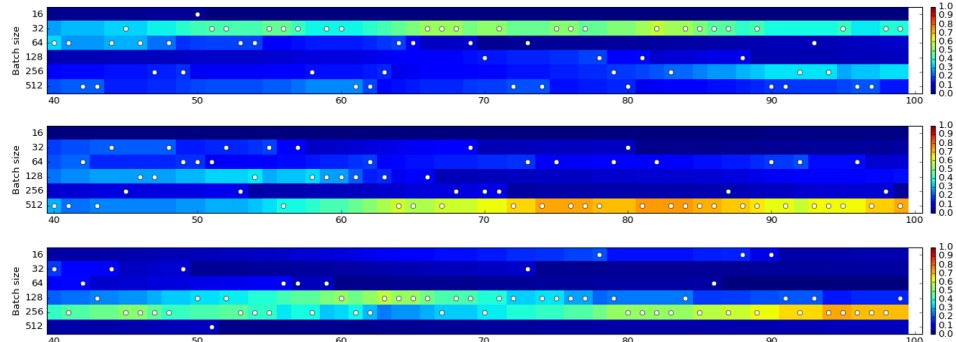

Figure 3: The probability distribution and selected batch size. The white dot is selected batch size at epoch. (top) The case that small batch size performs better. (middle) The case that large batch size performs better. (bottom) The case that best performing batch size varies.

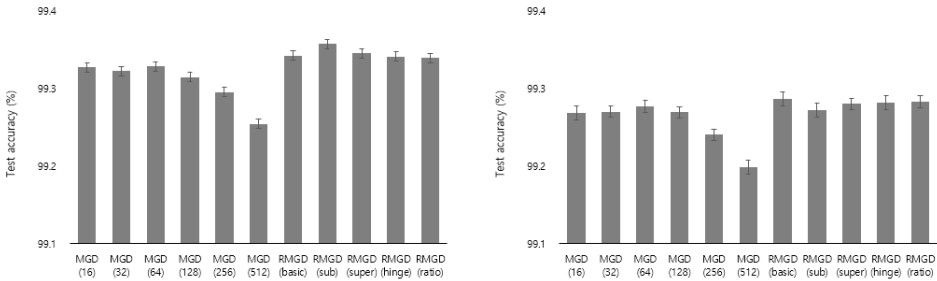

Figure 4: The results of test accuracy for the MNIST dataset. The error bar is standard error. (left) The test accuracy of 100 times repeated experiments with AdamOptimizer. (right) The test accuracy of 100 times repeated experiments with AdagradOptimizer. In both cases, most RMGD settings outperform all fixed MGD algorithms.

## 5.1 MNIST DATASET

The validity of the RMGD was assessed by performing image classification on the MNIST dataset using AdamOptimizer and AdagradOptimizer as optimizer. The experiments were repeated 100 times for each algorithm and each optimizer, then the results were analyzed for significance. Figure 3 shows the probability distribution and the selected batch size with respect to epoch during training for the RMGD. The white dot represents the batch size selected at each epoch. The top figure is the case that small batch size (32) performs better. After epoch 50, batch size 32 gets high probability and is selected more than others. It means that batch size 32 has less misupdating in this case. The gradually increasing batch size algorithm may not perform well in this case. The middle figure is the case that large batch size (512) performs better. After epoch 60, batch size 512 gets high probability and selected more than others. The bottom figure shows that the best performing batch size varies with epoch. During epoch from 40 to 55, batch size of 256 performs best, and best performing batch size switches to 128 during epoch from 60 to 70, then better performing batch size backs to 256 after epoch 80. In the results, any batch size can be a successful batch size in the later stages without any particular order. The RMGD is more flexible for such situation than the MGD or directional adaptive MGD such as gradually increasing batch size algorithm.

Figure 4 shows the test accuracy of each algorithm. The error bar is standard error. The number in parenthesis next to MGD represents the batch size used in the MGD. 'Basic', 'sub', 'super', 'hinge', and 'ratio' in parenthesis next to RMGD represent RMGD settings 'batch size set equal to grid search, 0-1 loss', 'subset of basic, 0-1 loss', 'superset of basic, 0-1 loss', 'basic set, hinge loss', and 'basic set, percentage of non-negative changes in validation loss', respectively. The left figure is the test accuracy with AdamOptimizer. The right figure is the test accuracy with AdagradOptimizer. Among the MGD algorithms, relatively small batch sizes (16 - 64) lead to higher performance than large batch sizes (128 - 512) and batch size 64 achieves the best performance in grid search. These

Table 1: Iterations and real time for training, and test accuracy of MNIST classification with AdamOptimizer. The 'total' is the sum of the average values from MGD 16 to 512, which means the whole grid search is performed.

| Algorithms | Iterations | Real time (sec) | Test accuracy (%) | | |
|---|---|---|---|---|---|
| | | | Mean ± SD | Max | Min |
| MGD (16) | 343,800 | 1,221.54 ± 36.00 | 99.327 ± 0.064 | 99.480 | 99.140 |
| MGD (32) | 171,900 | 697.82 ± 19.70 | 99.322 ± 0.060 | 99.500 | 99.150 |
| MGD (64) | 86,000 | 379.14 ± 11.32 | 99.328 ± 0.058 | 99.460 | 99.170 |
| MGD (128) | 43,000 | 262.33 ± 2.34 | 99.314 ± 0.056 | 99.440 | 99.170 |
| MGD (256) | 21,500 | 208.13 ± 2.20 | 99.295 ± 0.059 | 99.470 | 99.170 |
| MGD (512) | 10,800 | 180.06 ± 0.37 | 99.254 ± 0.054 | 99.430 | 99.110 |
| MGD (total) | 677,000 | 2,949.02 | | | |
| RMGD (basic) | 68,309 ± 8,900 | 333.73 ± 25.38 | 99.342 ± 0.064 | 99.480 | 99.110 |
| RMGD (sub) | 85,777 ± 12,112 | 400.73 ± 51.91 | **99.357 ± 0.057** | 99.510 | 99.060 |
| RMGD (super) | 67,948 ± 6,022 | 332.61 ± 22.26 | 99.345 ± 0.058 | 99.460 | 99.110 |
| RMGD (hinge) | 69,607 ± 8,887 | 337.38 ± 25.29 | 99.341 ± 0.062 | 99.480 | 99.130 |
| RMGD (ratio) | 95,530 ± 8,281 | 449.37 ± 26.71 | 99.339 ± 0.062 | 99.470 | 99.150 |

Table 2: Iterations and real time for training, and test accuracy of MNIST classification with AdagradOptimizer.

| Algorithms | Iterations | Real time (sec) | Test accuracy (%) | | |
|---|---|---|---|---|---|
| | | | Mean ± SD | Max | Min |
| MGD (16) | 343,800 | 1,160.87 ± 22.34 | 99.268 ± 0.090 | 99.430 | 98.920 |
| MGD (32) | 171,900 | 640.68 ± 15.53 | 99.270 ± 0.070 | 99.410 | 99.050 |
| MGD (64) | 86,000 | 367.40 ± 12.63 | 99.277 ± 0.077 | 99.440 | 99.110 |
| MGD (128) | 43,000 | 262.48 ± 1.37 | 99.269 ± 0.069 | 99.410 | 99.080 |
| MGD (256) | 21,500 | 195.60 ± 2.00 | 99.240 ± 0.072 | 99.390 | 99.030 |
| MGD (512) | 10,800 | 170.31 ± 1.41 | 99.198 ± 0.085 | 99.390 | 98.810 |
| MGD (total) | 677,000 | 2,797.34 | | | |
| RMGD (basic) | 68,159 ± 8,447 | 323.33 ± 23.57 | **99.286 ± 0.088** | 99.490 | 98.900 |
| RMGD (sub) | 81,479 ± 9,141 | 356.16 ± 28.06 | 99.272 ± 0.092 | 99.460 | 98.960 |
| RMGD (super) | 67,638 ± 7,733 | 320.38 ± 23.55 | 99.280 ± 0.074 | 99.410 | 99.090 |
| RMGD (hinge) | 68,199 ± 8,642 | 322.33 ± 24.03 | 99.282 ± 0.089 | 99.420 | 98.900 |
| RMGD (ratio) | 93,523 ± 9,871 | 452.69 ± 34.71 | 99.283 ± 0.078 | 99.480 | 99.020 |

results correspond with other studies Breuel (2015); Keskar et al. (2016); Mishkin et al. (2017). Most RMGD settings outperform all fixed MGD algorithms in both case. Although the performance of RMGD is not significantly increased compared to the best MGD, the purpose of this algorithm is not to improve performance, but to ensure that the best performance is achieved without performing a grid search on the batch size. Rather, the improved performance of the RMGD is a surprising result. Therefore, the RMGD is said to be valid. There are little performance gap among RMGD settings. The 'sub' setting outperforms the 'basic' setting in left figure, but the opposite result is shown in right figure. Therefore, there is no clear tendency of performance change depending on the size of the batch size set.

Table 1 and 2 present iterations and real time for training, mean, maximum, and minimum of test accuracies for each algorithm with AdamOptimizer and AdagradOptimizer respectively. The MGD (total) is the summation of the iterations and real time of whole MGDs for grid search. The RMGD (basic) outperforms best performing MGD and is, also, faster than best performing MGD. Furthermore, it is 8 times faster than grid search in both cases. In the results, the RMGD is effective regardless of the optimizer.

## 5.2 CIFAR10 AND CIFAR100 DATASET

The CIFAR10 and CIFAR100 dataset were, also, used to assess effectiveness of the RMGD. The experiments were repeated 25 times and 10 times, respectively. In these experiments, all images

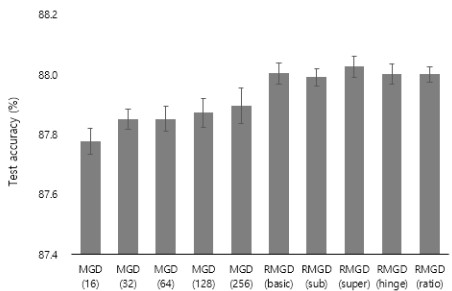 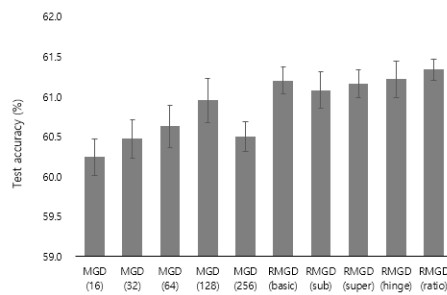

Figure 5: The results of test accuracy for the CIFAR10 and CIFAR100 dataset. The error bar is standard error. (left) The test accuracy of 25 times repeated experiments on CIFAR10. (right) The test accuracy of 10 times repeated experiments on CIFAR100. In both cases, all RMGD settings outperform all fixed MGD algorithms.

Table 3: Iterations and real time for training, and test accuracy on CIFAR10.

| Algorithms | Iterations | Real time (sec) | Test accuracy (%) | | |
|---|---|---|---|---|---|
| | | | Mean ± SD | Max | Min |
| MGD (16) | 1,072,050 | 10,085.26 ± 216.48 | 87.778 ± 0.207 | 88.290 | 87.480 |
| MGD (32) | 536,200 | 7,643.93 ± 459.95 | 87.851 ± 0.160 | 88.250 | 87.630 |
| MGD (64) | 268,100 | 6,160.16 ± 68.54 | 87.853 ± 0.202 | 88.330 | 87.450 |
| MGD (128) | 134,050 | 5,675.15 ± 181.80 | 87.873 ± 0.234 | 88.210 | 87.090 |
| MGD (256) | 67,200 | 5,466.79 ± 402.20 | 87.897 ± 0.293 | 88.260 | 87.170 |
| MGD (total) | 2,077,600 | 35,031.29 | | | |
| RMGD (basic) | 463,629 ± 48,692 | 7,592.43 ± 403.65 | 88.004 ± 0.167 | 88.380 | 87.780 |
| RMGD (sub) | 507,186 ± 93,961 | 7,614.16 ± 514.20 | 87.992 ± 0.147 | 88.270 | 87.730 |
| RMGD (super) | 397,685 ± 33,535 | 7,426.11 ± 228.28 | **88.027 ± 0.179** | 88.340 | 87.760 |
| RMGD (hinge) | 459,664 ± 56,086 | 7,584.01 ± 439.62 | 88.003 ± 0.167 | 88.380 | 87.810 |
| RMGD (ratio) | 426,123 ± 15,213 | 7,561.72 ± 220.56 | 88.002 ± 0.129 | 88.330 | 87.770 |

Table 4: Iterations and real time for training, and test accuracy on CIFAR100.

| Algorithms | Iterations | Real time (sec) | Test accuracy (%) | | |
|---|---|---|---|---|---|
| | | | Mean ± SD | Max | Min |
| MGD (16) | 1,072,050 | 12,097.47 ± 57.47 | 60.247 ± 0.690 | 61.940 | 59.620 |
| MGD (32) | 536,200 | 8,058.14 ± 39.87 | 60.475 ± 0.721 | 61.750 | 59.290 |
| MGD (64) | 268,100 | 6,400.21 ± 12.78 | 60.628 ± 0.795 | 61.950 | 59.170 |
| MGD (128) | 134,050 | 5,598.85 ± 38.18 | 60.954 ± 0.834 | 62.120 | 59.530 |
| MGD (256) | 67,200 | 5,245.88 ± 40.67 | 60.504 ± 0.553 | 61.560 | 59.830 |
| MGD (total) | 2,077,600 | 37,400.55 | | | |
| RMGD (basic) | 425,416 ± 44,392 | 7,503.09 ± 281.79 | 61.203 ± 0.502 | 62.050 | 60.310 |
| RMGD (sub) | 532,624 ± 69,195 | 7,841.88 ± 530.29 | 61.080 ± 0.720 | 61.910 | 59.900 |
| RMGD (super) | 397,717 ± 20,091 | 7,408.00 ± 163.74 | 61.166 ± 0.560 | 61.970 | 60.320 |
| RMGD (hinge) | 419,100 ± 53,491 | 7,476.71 ± 324.29 | 61.219 ± 0.714 | 62.060 | 59.580 |
| RMGD (ratio) | 412,532 ± 15,660 | 7,456.50 ± 100.79 | **61.340 ± 0.411** | 61.880 | 60.550 |

are whitened and contrast normalized before being input to the network. Figure 5 shows the test accuracy for each algorithm. The left figure represents the test accuracy on CIFAR10. In contrast to the MNIST results, relatively large batch sizes (128 - 256) lead to higher performance than small batch sizes (16 - 64) and batch size 256 achieves the best performance in grid search. The right figure represents the test accuracy on CIFAR100 and batch size 128 achieves the best performance in grid search. The results on MNIST, CIFAR10 and CIFAR100 indicate that it is difficult to know which batch size is optimal before performing a grid search. Meanwhile, all RMGD settings have again exceeded the best performance of fixed MGD. There are no significant performance gaps among RMGD settings, so there is no need to worry about choosing appropriate batch size set or selecting cost function.

Table 3 and 4 present the detailed results on CIFAR10 and CIFAR100 dataset. The RMGD (basic) is a little slower than single best performing MGD (256 for CIFAR10 and 128 for CIFAR100), how-ever, it was much faster than grid search -about 4.6 times on CIFAR10 and 5.0 times on CIFAR100 faster. Therefore, this results, also, show the effectiveness of the RMGD.

It is difficult to compare the RMGD with other adaptive batch size algorithm, e.g. coupling adaptive batch sizes (CABS) Balles et al. (2017), directly since the underlying goals are different. While the goal of the RMGD is to reduce the validation loss in terms of generalization performance, the CABS determines the batch size to balance between the gradient variance and computation. However, it is obvious that the RMGD is simpler and easier to implement than any other adaptive algorithm cited in this paper, and comparing the test accuracy between the RMGD and the CABS on the CIFAR10 and CIFAR100 using the same experimental settings with 'All-CNN-C' shows that the performance of the RMGD is higher than that of the CABS (CIFAR10: $87.862 \pm 0.142$, CIFAR100: $60.782 \pm 0.421$). And again, the purpose of this algorithm is not to outperform other algorithms, but to guarantee that the best performance is reached without grid search.

## CONCLUSION

Selecting batch size affects the model quality and training efficiency, and determining the appro-priate batch size is time consuming and requires considerable resources as it often relies on grid search. The focus of this paper is to design a simple robust algorithm that is theoretically sound and applicable in many situations.

This paper considers a resizable mini-batch gradient descent (RMGD) algorithm based on a multi-armed bandit that achieves equivalent performance to that of best fixed batch-size. At each epoch, the RMGD samples a batch size according to certain probability distribution of a batch being successful in reducing the loss function. Sampling from this probability provides a mechanism for exploring different batch size and exploiting batch sizes with history of success. After obtaining the validation loss at each epoch with the sampled batch size, the probability distribution is updated to incorporate the effectiveness of the sampled batch size.

The goal of this algorithm is not to achieve state-of-the-art accuracy but rather to select appropri-ate batch size which leads low misupdating and performs better. The RMGD essentially assists the learning process to explore the possible domain of the batch size and exploit successful batch size. The benefit of RMGD is that it avoids the need for cumbersome grid search to achieve best performance and that it is simple enough to apply to various field of machine learning including deep learning using MGD. Experimental results show that the RMGD achieves the best grid search performance on various dataset, networks, and optimizers. Furthermore, it, obviously, attains this performance in a shorter amount of time than the grid search. Also, there is no need to worry about which batch size set or cost function to choose when setting RMGD. In conclusion, the RMGD is effective and flexible mini-batch gradient descent algorithm.

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

## APPENDIX

## A    REGRET BOUND

In the RMGD algorithm, there are $K$ batch sizes as multi arms with the probability distribution $\boldsymbol{\pi} \in S$, and at each epoch the algorithm should select one of the batch sizes $b_{k_\tau}$. Then it receives a cost of selecting this arm, $y_\tau^{k_\tau} \in \{0, 1\}$ by testing the validation loss $\ell$. The vector $\boldsymbol{y}_\tau \in \{0, 1\}^K$ represents the selecting cost for each batch size. The goal of this algorithm is to have low regret for not selecting the best performing batch size.

$$\text{Regret}_{\mathcal{T}}(S) = \mathbb{E}\left[\sum_{\tau=1}^{\mathcal{T}} y_\tau^{k_\tau}\right] - \min_i \sum_{\tau=1}^{\mathcal{T}} y_\tau^i \tag{3}$$

where the expectation is over the algorithm's randomness of batch size selection.

Let $S$ be the probability simplex, the selecting loss functions be $f_\tau(\boldsymbol{\pi}) = \langle \boldsymbol{\pi}, \boldsymbol{y}_\tau \rangle^2$ and $R : S \to \mathbb{R}$ be a regularization function that is often chosen to be strongly convex with respect to some norm $\|\cdot\|$. The algorithm select a batch size with probability $\mathbb{P}[b_{k_\tau}] = \pi_\tau^{k_\tau}$ and therefore $f_\tau(\boldsymbol{\pi}_\tau)$ is the expected cost of the selected batch size at epoch $\tau$. The gradient of the selecting loss function is $\boldsymbol{y}_\tau$. However, only one element $y_\tau^{k_\tau}$ is known at each epoch. To estimate gradient, random vector $\boldsymbol{z}_\tau$ is defined as follows:

$$z_\tau^i = \begin{cases} y_\tau^i / \pi_\tau^i & \text{if } i = k_\tau \\ 0 & \text{otherwise} \end{cases}$$

and expectation of $\boldsymbol{z}_\tau$ satisfies,

$$\mathbb{E}[\boldsymbol{z}_\tau | \boldsymbol{z}_{\tau-1}, \ldots, \boldsymbol{z}_0] = \sum_{i=1}^{K} \mathbb{P}[b_{k_\tau}] z_\tau^i = \pi_\tau^{k_\tau} \frac{y_\tau^{k_\tau}}{\pi_\tau^{k_\tau}} = y_\tau^{k_\tau}.$$

The most natural learning rule is to set the probability distribution which has minimal cost on all past epochs. It is referred to as Follow-the-Regularized-Leader (FTRL) in online learning:

$$\forall \tau, \quad \boldsymbol{\pi}_{\tau+1} = \underset{\boldsymbol{\pi} \in S}{\arg\min} \left\{ \beta \sum_{t=1}^{\tau} f_t(\boldsymbol{\pi}) + R(\boldsymbol{\pi}) \right\},$$

where $\beta$ is positive hyperparameter. The FTRL has a problem that it requires solving an optimization problem at each epoch. To solve this problem, Online Mirror Descent (OMD) is applied. The OMD computes the current probability distribution iteratively based on a gradient update rule and the previous probability distribution and lies in the update being carried out in a 'dual' space, defined by regularizer. This follows from considering $\nabla R$ as a mapping from $\mathbb{R}^K$ onto itself. The OMD relies on Bregman divergence. The Bregman divergence between $\boldsymbol{\pi}$ and $\tilde{\boldsymbol{\pi}}$ with respect to the regularizer $R$ is given as:

$$B_R(\boldsymbol{\pi} \| \tilde{\boldsymbol{\pi}}) = R(\boldsymbol{\pi}) - R(\tilde{\boldsymbol{\pi}}) - \nabla R(\tilde{\boldsymbol{\pi}}) \cdot (\boldsymbol{\pi} - \tilde{\boldsymbol{\pi}}), \tag{4}$$

and a Bregman projection of $\tilde{\boldsymbol{\pi}}$ onto simplex $S$:

$$\underset{\boldsymbol{\pi} \in S}{\arg\min} \, B_R(\boldsymbol{\pi} \| \tilde{\boldsymbol{\pi}}).$$

Then the probability distribution is updated by the OMD as follows:

$$\begin{aligned} \nabla R(\tilde{\boldsymbol{\pi}}_{\tau+1}) &= \nabla R(\tilde{\boldsymbol{\pi}}_\tau) - \beta \boldsymbol{z}_\tau \\ \boldsymbol{\pi}_{\tau+1} &= \underset{\boldsymbol{\pi} \in S}{\arg\min} \, B_R(\boldsymbol{\pi} \| \tilde{\boldsymbol{\pi}}_{\tau+1}). \end{aligned}$$

In general, if $R$ is strongly convex, then $\nabla R$ becomes a bijective mapping, thus $\tilde{\boldsymbol{\pi}}_{\tau+1}$ can be recovered by the inverse gradient mapping $(\nabla R)^{-1}$. Given that $R$ is strongly convex, the OMD and FTRL produce equivalent predictions:

$$\underset{\boldsymbol{\pi} \in S}{\arg\min} \, B_R(\boldsymbol{\pi} \| \tilde{\boldsymbol{\pi}}_{\tau+1}) = \underset{\boldsymbol{\pi} \in S}{\arg\min} \left\{ \beta \sum_{t=1}^{\tau} f_t(\boldsymbol{\pi}) + R(\boldsymbol{\pi}) \right\}$$

---

[2] $\langle \boldsymbol{\pi}, \boldsymbol{y} \rangle$ is the inner product between vectors $\boldsymbol{\pi}$ and $\boldsymbol{y}$

by the Lemma 1 in Hazan & Kale (2010). It makes sense to use the negative entropic regularization for $R$ in RMGD setting:

$$R(\boldsymbol{\pi}) = \sum_{i=1}^{K} \pi^i \log(\pi^i).$$

Then, $\nabla R(\boldsymbol{\pi})_i = \log(\pi^i) + 1$. From the OMD, $\tilde{\boldsymbol{\pi}}_{\tau+1}$ is updated as follows:

$$
\begin{aligned}
\nabla R(\tilde{\boldsymbol{\pi}}_{\tau+1}) &= \nabla R(\tilde{\boldsymbol{\pi}}_\tau) - \beta \boldsymbol{z}_\tau \\
\log(\tilde{\pi}^i_{\tau+1}) + 1 &= \log(\tilde{\pi}^i_\tau) + 1 - \beta z^i_\tau \\
\tilde{\pi}^i_{\tau+1} &= \tilde{\pi}^i_\tau e^{-\beta z^i_\tau}.
\end{aligned}
$$

The Bregman projection with respect to the negative entropy function becomes scaling by the $\ell_1$-norm. Therefore,

$$\pi^i_{\tau+1} = \frac{\tilde{\pi}^i_{\tau+1}}{\sum_j \tilde{\pi}^j_{\tau+1}}. \tag{5}$$

The probability distribution $\boldsymbol{\pi}_\tau$ is updated by the rule of the normalized exponentiated gradient (normalized-EG) algorithm described in Algorithm 1. Also, the selecting loss function is linear and it is satisfied that $\forall \tau, i$ we have $\beta z^i_\tau \geq 0$. Then,

$$\sum_{\tau=1}^{\mathcal{T}} \langle \boldsymbol{\pi}_\tau - \boldsymbol{\pi}^*, \boldsymbol{z}_\tau \rangle \leq \frac{\log(K)}{\beta} + \beta \sum_{\tau=1}^{\mathcal{T}} \sum_{i=1}^{K} \pi^i_\tau (z^i_\tau)^2 \tag{6}$$

by the Theorem 2.22 in Shalev-Shwartz et al. (2012), where $\boldsymbol{\pi}^* \in S$ is a fixed vector which minimizes the cumulative selecting loss,

$$\boldsymbol{\pi}^* = \arg\min_{\boldsymbol{\pi} \in S} \sum_{\tau=1}^{\mathcal{T}} f_\tau(\boldsymbol{\pi}).$$

Since $f_\tau$ is convex and $\boldsymbol{z}_\tau$ is estimated gradients for all $\tau$,

$$\mathbb{E}\left[\sum_{\tau=1}^{\mathcal{T}} (f_\tau(\boldsymbol{\pi}_\tau) - f_\tau(\boldsymbol{\pi}^*))\right] \leq \frac{\log(K)}{\beta} + \beta \sum_{\tau=1}^{\mathcal{T}} \mathbb{E}\left[\sum_{i=1}^{K} \pi^i_\tau (z^i_\tau)^2\right] \tag{7}$$

by the Theorem 4.1 in Shalev-Shwartz et al. (2012). The last term is bounded as follows:

$$
\begin{aligned}
\mathbb{E}\left[\sum_{i=1}^{K} \pi^i_\tau (z^i_\tau)^2\right] &= \sum_{j=1}^{K} \mathbb{P}[k_\tau = j] \sum_{i=1}^{K} \pi^i_\tau (z^i_\tau)^2 \\
&= \sum_{j=1}^{K} (\pi^j_\tau)^2 (y^j_\tau / \pi^j_\tau)^2 \\
&= \sum_{j=1}^{K} (y^j_\tau)^2 \leq K.
\end{aligned}
\tag{8}
$$

Therefore, the regret of the RMGD is bounded,

$$\mathbb{E}\left[\sum_{\tau=1}^{\mathcal{T}} y^{k_\tau}_\tau\right] - \min_i \sum_{\tau=1}^{\mathcal{T}} y^i_\tau \leq \frac{\log K}{\beta} + \beta K \mathcal{T}. \tag{9}$$

In particular, setting $\beta = \sqrt{\log(K)/(K\mathcal{T})}$, the regret is bounded by $2\sqrt{K \log(K)\mathcal{T}}$, which is sublinear with $\mathcal{T}$.

## B  EXPERIMENTAL SETTINGS

### DATASET

**MNIST** is a dataset of handwritten digits that is commonly used for image classification. Each sample is a black and white image and $28 \times 28$ in size. The MNIST is split into three parts: 55,000 samples for training, 5,000 samples for validation, and 10,000 samples for test.

**CIFAR10** consists of 60,000 $32 \times 32$ color images in 10 classes (airplane, automobile, bird, cat, deer, dog, frog, horse, ship, and truck), with 6,000 images per class. The CIFAR10 is split into three parts: 45,000 samples for training, 5,000 samples for validation, and 10,000 samples for test.

**CIFAR100** consists of 60,000 $32 \times 32$ color images in 100 classes. The CIFAR100 is split into three parts: 45,000 samples for training, 5,000 samples for validation, and 10,000 samples for test.

### SETTINGS

The simple CNN consists of two convolution layers with $5 \times 5$ filter and $1 \times 1$ stride, two max pooling layers with $2 \times 2$ kernel and $2 \times 2$ stride, single fully-connected layer, and softmax classifier. Description of the 'All-CNN-C' is provided in Table 5. For MNIST, AdamOptimizer with $\eta = 10^{-4}$ and AdagradOptimizer with $\eta = 0.1$ are used as optimizer. The basic batch size set $\mathcal{B} = \{16, 32, 64, 128, 256, 512\}$, subset of basic $\mathcal{B}^- = \{16, 64, 256\}$, and superset of basic $\mathcal{B}^+ = \{16, 24, 32, 48, 64, 96, 128, 192, 256, 384, 512\}$. The model is trained for a total of 100 epochs. For CIFAR10 and CIFAR100, MomentumOptimizer with fixed momentum of 0.9 is used as optimizer. The learning rate $\eta^k$ is scaled up proportionately to the batch size ($\eta^k = 0.05 * b_k/256$) and decayed by a schedule $S = [200, 250, 300]$ in which $\eta^k$ is multiplied by a fixed multiplier of 0.1 after 200, 250, and 300 epochs respectively. The model is trained for a total of 350 epochs. Dropout is applied to the input image as well as after each convolution layer with stride 2. The dropout probabilities are 20% for dropping out inputs and 50% otherwise. The model is regularized with weight decay $\lambda = 0.001$. The basic batch size set $\mathcal{B} = \{16, 32, 62, 128, 256\}$, subset of basic $\mathcal{B}^- = \{16, 64, 256\}$, and superset of basic $\mathcal{B}^+ = \{16, 24, 32, 48, 64, 96, 128, 192, 256\}$. For all experiments, rectified linear unit (ReLU) is used as activation function. For RMGD, $\beta$ is set to $\sqrt{\log(6)/(6 * 100)} \approx 0.055$ for MNIST and $\sqrt{\log(5)/(5 * 350)} \approx 0.030$ for CIFAR10 and CIFAR100. The basic batch size selecting cost is 0-1 loss, hinge loss is $\max\{0, \ell_\tau - \ell_{\tau-1}\}$, and ratio loss is $\max\{0, (\ell_\tau - \ell_{\tau-1})/\ell_{\tau-1}\}$.

Table 5: Architecture of the All-CNN-C for CIFAR10 and CIFAR100

| Layer | Layer description |
|---|---|
| input | Input $32 \times 32$ RGB image |
| conv1 | $3 \times 3$ conv. 96 ReLU, stride 1, dropout 0.2 |
| conv2 | $3 \times 3$ conv. 96 ReLU, stride 1 |
| conv3 | $3 \times 3$ conv. 96 ReLU, stride 2 |
| conv4 | $3 \times 3$ conv. 192 ReLU, stride 1, dropout 0.5 |
| conv5 | $3 \times 3$ conv. 192 ReLU, stride 1 |
| conv6 | $3 \times 3$ conv. 192 ReLU, stride 2 |
| conv7 | $3 \times 3$ conv. 192 ReLU, stride 1, dropout 0.5 |
| conv8 | $1 \times 1$ conv. 192 ReLU, stride 1 |
| conv9 | $1 \times 1$ conv. 10 or 100 ReLU, stride 1 |
| pool | averaging over $6 \times 6$ spatial dimensions |
| softmax | 10-way or 100-way softmax |

