# OpenReview forum: "A Resizable Mini-batch Gradient Descent based on a Multi-Armed Bandit"
_ICLR.cc/2019/Conference_

### Official Review · AnonReviewer2 · 2018-11-01
**The contribution is incremental due to lack of originality and somewhat narrow scope of the application**

**Rating:** 4
**Confidence:** 4

**Review:**

The paper applies multi-armed bandits for choosing the size of the minibatch to be used in each training epoch of a standard CNN. The loss of the bandit is binary: zero if the validation loss decreases and 1 otherwise. In the experiments, the Exp3 bandit algorithm is run with Adam and Adagrad on MNIST, CIFAR-10, and CIFAR-100. The results show that the bandit approach allows to obtain a test error better (although not significantly better) than the test error corresponding to the best minibatch size among those considered by the bandit.

The idea of viewing the choice of hyperparameters in a learning algorithm as a bandit problem is known and has been explored in different contexts, although the specific application to minibatch size is new as far as I know.

The paper could have gained strength if bandits had been considered in wider context of parameter/model selection in deep learning.

It is not clear how results scale with the number and choice of the grid values.

I would have liked to see a more thorough investigation of the impact of the bandit loss on the experiments. It is true that as far as the theory is concerned, any bounded loss is OK. But I practice I would expect that a graduated loss (e.g., signed percentage of change in validation loss), would be more informative.

---

> ### Author Response · Authors · 2018-11-26
> **As a result of additional experiments, there were no significant performance gap according to the number of grid values or bandit loss.**
>
>
> 1. About bandit-based hyperparameter optimization and wider context of parameter/model selection:
>   First, the authors were shortsighted to consider batch size problem as a hyperparameter optimization problem, and for this problem, the proposed algorithm does not provide the best batch size instead it provides a policy (of adaptively sampling for different batch size) to attain performance which is obtained by using a best fixed batch size. It is not clear whether the same strategy will be applicable for other hyperparameters. Rather than giving the best possible hyperparameter through certain strategy as in previously proposed bandit-based hyperparameter optimization methods, this algorithm adaptively samples the batch size with each epoch and evaluates its performance on the validation data in order to update the statistics of the batch size. The accumulated performance is compared to the best performing hyperparameter’s accumulated performance.
>
> 2. The number and choice of the grid values and the bandit loss:
>   The authors are most grateful for the questions regarding the number of the grid values and the bandit loss. The authors conducted additional experiments by varying the number of grid values (reduced subset, extended superset) and using graduated loss (hinge, ratio) on the three different datasets: MNIST, CIFAR10, and CIFAR100. However, additional grid search with either reduced or extended grid set was not performed.
>   As ratio loss, the signed percentage of change in validation loss was considered. The probability of initially sampled batch size increases as validation loss decreased. In general, the validation loss decreases in the beginning of training regardless of which batch size sampled. The probability of the initially sampled batch size continued to rise, and started to hinder exploration from functioning normally. For this reason, the authors considered only percentage of non-negative changes in validation loss.
>   Interestingly, there are a little gap in performance, but they are not significant and there is no clear trend. Detailed experimental settings and results are described in the revised paper.

---

### Official Review · AnonReviewer3 · 2018-11-01
**Interesting extension of minibatch GD using a straight-forward application of bandits**

**Rating:** 7
**Confidence:** 4

**Review:**

This paper considers a resizable mini-batch gradient descent (RMGD) algorithm based on a multi-armed bandit for achieving best performance in grid search by selecting an appropriate batch size at each epoch with a probability defined as a function of its previous success/failure. Its results suggest that RMGD faster than MGD with grid search, and generalizes better.

The paper is well written. The idea itself is a simple and relatively straightforward application of bandits. The paper has some merits as it proposes an efficient and theoretically sound method to replace grid search in MGD.

One result that stands out is that RMGD achieves better results than the best performing batch size. The authors may want to discuss this in more depth. This may be due to the fact that the problem is inherently contextual: each epoch is different from other epochs, and may require a better-suited bach-size. Maybe contextual bandits would be a good candidate to try.

Comments:
- offer some analysis or explanation of the surprising results
- add equation numbers for ease of reference
- in 4.1, why did you use this particular probability update? motivate/explain this choice.
- appendix A: Specify that <> is dot product.
                        introduce Beta
                        briefly explain mirror descent
                        why is beta z >= -1? My sense is that it is >= 0. can it be negative?
                        explain, motivate or cite the equation following beta z >= -1

I am pretty familiar wit bandit literature. Less so with GD literature. The paper's hybrid approach, although simple, exposes interesting questions. I tend towards accepting the paper.

---

> ### Author Response · Authors · 2018-11-26
> **The details about OMD and probability update rule were described in revised paper.**
>
> The authors would like thank the reviewer for his/her detailed comments.
>
> 1. Some explanation of the surprising results:
>   RMGD samples the batch size every epoch to minimize the cumulative cost relative to that of the best fixed batch size. The cost (the performance of the sampled batch size) is measured in terms of increased validation loss. Thus, the RMGD attempts to dynamically select batch sizes that minimize the number of increase validation loss compared to that of the best fixed batch size. Minimizing the number of increase in validation loss during training will lead to small overall validation loss. However, this point was not explicitly conveyed in submitted manuscript. Although RMGD outperformed the best fixed batch size in all the experiments, it is not the intention of this algorithm: the algorithm only tries to minimize the regret and perform as well as the best. The authors believe that the theoretical and in-depth analysis of why RMGD outperformed the best performance of grid search is a worthwhile topic in future study.
>
> 2. Motivation of probability update rule:
>   \pi(π) lies in the probability simplex and a strongly convex regularizer offers a tighter regret bound in the follow-the-regularized-leader (FTRL) framework, so the RMGD uses negative entropic regularization function. Online mirror descent (OMD) is applied instead of FTRL to update the probability distribution. The OMD relies on Bregman divergence and projection, and if regularizer is strongly convex, then the OMD and FTRL produce equivalent predictions for each epoch. When the regularizer is negative entropic function, the probability distribution is updated as randomized weighted majority algorithm. The details are described in 'Appendix A' of the revised paper.
>
> 3. Explanation of online mirror descent:
>   The FTRL has a problem that it requires solving an optimization problem at each epoch. To solve this problem, OMD is applied. The OMD computes the current decision iteratively based on a gradient update rule and the previous decision, and lies in the update being carried out in a ‘dual’ space defined by regularizer. This follows from considering the gradient of the regularizer. In terms of implementation, the current probability is updated by the gradient update in the ‘dual’ space (regularizer’s gradient space) and Bregman projection onto convex set (simplex S). The OMD achieves the same regret bound as FTRL. The details are described in 'Appendix A' of the revised paper.
>
> 4. \beta z >= -1?:
>   In this algorithm, \beta z >= 0 as pointed out by the reviewer. By the Theorem 2.22 in Shalev-Shwartz et al. (2012), the equation (6) in 'Appendix A' is derived for \beta z ≥ -1, and this algorithm obviously satisfies this condition. The authors modified the manuscript to make this point clearer.
>
> 5. Citation of the equation following \beta z >= -1:
>   The equation (6) and (7) are derived by the Theorem 2.22 and 4.1 in Shalev-Shwartz et al. (2012). The reference is cited in the revised paper.

---

### Official Review · AnonReviewer1 · 2018-11-02
**Well-motivated idea seemingly showing a small improvement in performance, but at little cost and with a potential increase in robustness**

**Rating:** 6
**Confidence:** 3

**Review:**

The authors consider the problem of determining the minibatch size for SGD by first fixing a set of candidate sizes, and then learning a distribution over those sizes using a MAB algorithm. A minibatch size is first sampled from the distribution, then one training epoch is performed. A validation error is then computed, and if it is lower than that of the last epoch, the cost of the minibatch is taken to be zero (otherwise one), and the distribution is updated. This is Algorithm 1.

In Section 4.2, they prove a regret bound, but I don’t think that regret is really the correct notion, here (although it’s very close). This is a subtle point, so I’ll set up some notation. Let w(b_1, ..., b_t) be the result at the tth epoch, if the batch sizes b_1, …, b_t were used at the 1st through tth epochs. Let y(w,b) be 0 if training one epoch starting at w with batch size b would improve the validation error, and 1 otherwise.

They show (unnumbered inequality on the middle of page 5) that \sum_t y(w(b_1,...,b_{t-1}),b_t) is close to \sum_t y(w(b_1,...,b_{t-1}),b^*), where b_t is the batch size that was chosen at time t, and b^* is the best fixed batch size. The key point here is that the comparator (the second sum) starts each epoch at the result that was found by their adaptive algorithm, *not* what would have been found if a batch size of b^* had been used from the beginning.

In other words, their result does *not* show that their algorithm is close to outperforming a fixed choice of batch size (for that to hold, the comparator would need to be \sum_t y(w(b^*,...,b^*),b^*)). What they show is similar, but subtly different. They don’t put too much weight on this theoretical result, and in fact don’t even explicitly claim that the comparator in this result is that for a fixed choice of batch size, so really this is a minor issue, but I think that this is something that should be clarified, since it would be easy for a reader to draw an incorrect conclusion.

With that said, their approach is well-motivated, and their experiments seem to show consistent small improvements in performance. I don’t think the performance improvements are totally conclusive, but one of the most appealing properties of their proposal is that it shouldn’t be much more computationally expensive than using a fixed minibatch size. Furthermore, their approach is potentially more robust, since you can presumably be less careful about choosing the set of candidate minibatch sizes, than you would be for choosing only one. So while the experiments don’t show a big improvement, their proposal has other benefits.

---

> ### Author Response · Authors · 2018-11-26
> **The comparator in this paper functions exactly as described by the reviewer.**
>
> The authors would like thank the reviewer for his/her insightful comment; however, we believe the manuscript may have misguided the reviewer into thinking that the comparator is adaptive but as the reviewer so rightly has pointed out the comparator must be fixed. The proposed RMGD does it exactly as described by the reviewer. The comparator in this paper functions exactly as described by the reviewer. In other words, a fixed i that minimizes cumulative sum is used throughout the run such that each epoch uses same i which represents the i-th batch in the batch set. Thus, the comparator is equivalent to \sum_t y(w(b^*, ..., b^*), b^*)). The authors have revised the manuscript to make this point more clear. We thank the reviewer for making this pointing.

---

### Author Response · Authors · 2018-11-26
**Summary of paper revision**

The followings were modified from the original paper.

Major revision:
  - Additional experimental settings and results for varying the number of grid values and using graduated bandit loss on three datasets (MNIST, CIFAR10, CIFAR100) were added in ‘5. Experiments and Appendix B’.
  - The description of online mirror descent (OMD) and the derivation of the probability update rule were described in ‘Appendix A’.

Minor revision:
  - We removed one reference [Bergstra & Bengio (2012)] not mentioned in this paper and added three references [Jamieson & Talwalkar (2016) and Li et al. (2017) in ‘2. Related works’ for bandit-based hyperparameter optimization, and Hazan & Kale (2010) in ‘Appendix A’ for the OMD theory].
  - We modified or added some sentences in ‘Abstract, 4.2 Regret bound, conclusion, and Appendix A’ to make more clear.
  - We added some equation numbers.

---

### Public Comment · (anonymous) · 2019-01-06
**Summary of submission to ICLR reproducibility challenge 2019**

We reproduce the results for this 2019 ICLR submission under the ICLR reproducibility challenge 2019. We also review the claims and findings of the paper carefully. We study the probability distributions which the algorithm converges to during different runs, and try to explain the behaviour. We point out some limitations in their work, provide a discussion of their results. We also try to verify the extent of their claims, and actual effect of their algorithm. Precisely, we try to find how much tuning does it require and how exactly is it benefiting the test accuracy. This underscores some limitations in their work. Finally, we offer some suggestions to improve their research. Our full report can be found at https://openreview.net/forum?id=SJlGaRBJMN . Our reproducibility code can be found at https://github.com/AnayMehrotra/Reproducibility_Challenge_ICLR_2019 .

---

### Meta-Review · Area_Chair1 · 2018-12-11
**Good simple idea but with limited novelty and gain of performance.**

**Confidence:** 4
**Recommendation:** Reject

**Metareview:**

It is a simple but good idea to consider the choice of mini-batch size as a multi-armed bandit problem. Experiments also show a slight improvement compared to the best fixed batch size.

The main concerns from the reviewers are that (1) treating the choice of hyper-parameters as a bandit problem is known and has been exploited in different context, and this paper is limited to the choice of the mini-batch size, (2) the improvement in the test error is not significant. The authors' feedback did not solve the concerns raised by R2.

This paper conveys a nice idea but as the current form it falls slightly below the standard of the ICLR publications. One direction for improvement, as suggested by the reviewer, would be extending the idea for a wider hyper-parameter selection problems.